# Proximal Graphical Event Models

**Debarun Bhattacharjya**      **Dharmashankar Subramanian**      **Tian Gao**

IBM Research

Thomas J. Watson Research Center, Yorktown Heights, NY, USA

{debarunb,dharmash,tgao}@us.ibm.com

## Abstract

Event datasets involve irregular occurrences of events over the timeline and are
prevalent in numerous domains. We introduce proximal graphical event models
(PGEMs) as a representation of such datasets. PGEMs belong to a broader family
of graphical models that characterize relationships between various types of events;
in a PGEM, the rate of occurrence of an event type depends only on whether or
not its parents have occurred in the most recent history. The main advantage over
state-of-the-art models is that learning is entirely data driven and without the need
for additional inputs from the user, which can require knowledge of the domain
such as choice of basis functions and hyper-parameters. We theoretically justify our
learning of parental sets and their optimal windows, proposing sound and complete
algorithms in terms of parent structure learning. We present efficient heuristics for
learning PGEMs from data, demonstrating their effectiveness on synthetic and real
datasets.

## 1   Introduction and Related Work

Event datasets are sequences of events of various types that typically occur as irregular and asyn-
chronous continuous-time arrivals. This is in contrast to time series data, which are observations of
continuous-valued variables over regular discrete epochs in time. Examples of event datasets include
logs, transactions, notifications and alarms, insurance claims, medical events, political events, and
financial events.

It is well known that a multivariate point process is able to capture the dynamics of events occurring
in continuous time, under reasonable regularity conditions, using *conditional intensity functions*.
These are akin to hazard rates in survival analysis and represent the rate at which an event type
occurs, conditioned on the history of event occurrences. Learning arbitrary history-dependent
intensity functions can be difficult and impractical, thus the literature makes various simplifying
assumptions. Some examples of such point processes include continuous time noisy-or (CT-NOR)
models [Simma et al., 2008], Poisson cascades [Simma and Jordan, 2010], Poisson networks [Rajaram
et al., 2005], piecewise-constant conditional intensity models [Gunawardana et al., 2011], forest-based
point processes [Weiss and Page, 2013], multivariate Hawkes processes [Zhou et al., 2013], and
non-homogeneous Poisson processes [Goulding et al., 2016].

*Graphical event models* (GEMs) have been proposed as a graphical representation for multivariate
point processes [Didelez, 2008, Meek, 2014, Gunawardana and Meek, 2016]. Unlike graphical
models for discrete-time dynamic uncertain variables such as dynamic Bayesian networks [Dean
and Kanazawa, 1989, Murphy, 2002] and time series graphs [Eichler, 1999, Dahlhaus, 2000], GEMs
capture continuous-time processes. They also differ from continuous-time Bayesian networks
[Nodelman et al., 2002], which represent homogeneous Markov models of the joint trajectories of
discrete variables rather than models of event streams in continuous time. GEMs provide a framework
that generalizes many of the afore-mentioned history-dependent models for event datasets, many of
which make the assumption of piece-wise constant conditional intensity functions. The literature

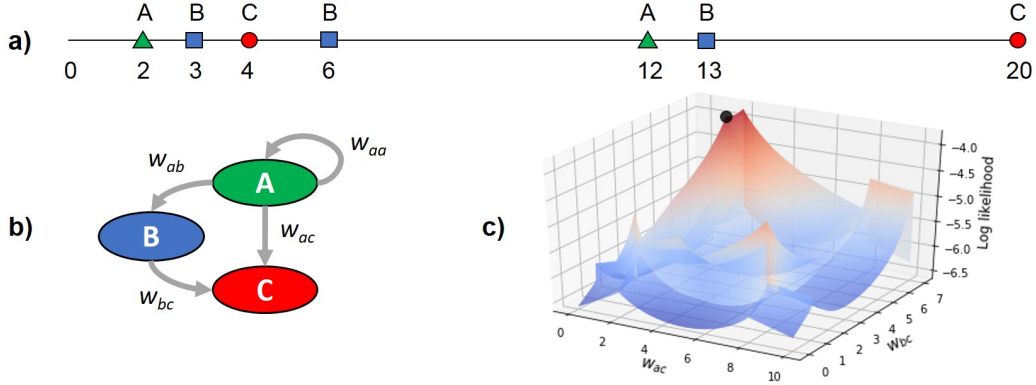

Figure 1: An example involving $M = 3$ event labels: a) Event dataset; b) Example PGEM; c) Surface plot of log likelihood for node $C$, given parents $A$ and $B$, as a function of windows $w_{ac}$ and $w_{bc}$.

takes varying approaches to the representation and learning of such functions, including decision trees [Gunawardana et al., 2011], forests [Weiss and Page, 2013], and generalized linear models [Rajaram et al., 2005].

A major drawback in these existing approaches is that they require the user to specify a set of basis functions in the form of relevant time intervals in the history. It is not obvious beforehand in most applications how to specify such a basis. Alternatively, if a basis is chosen in a manner that is exhaustively data-adaptive, i.e. using all historical epochs of event arrivals to define all historical time intervals of interest, one ends up with a prohibitively large basis set that makes any learning impractical. Thus there is a need to investigate approaches that don't require such a basis set specification and yet provide practical learning algorithms.

In this paper, we introduce *proximal graphical event models* (PGEMs), where the intensity of an event label depends on occurrences of its parent event labels in the graph within the most recent history, i.e. in temporal proximity (see Figure 1(b) for an example). Although PGEMs are a special case from the piece-wise constant conditional intensity family, there are several advantages of such models and our work on learning them from data: 1) They are practical models, capturing the assumption that the most recent history is sufficient for understanding how the future may unfold. We argue that PGEMs are particularly interpretable event models and could be useful for providing insights about the dynamics in an event dataset to political or financial analysts or medical practitioners or scientists; 2) Importantly, we present data-driven algorithms that learn a PGEM from an event dataset without additional user information, unlike the state-of-the-art models; 3) We present polynomial time heuristic algorithms that make PGEM learning computationally more tractable and therefore amenable to large event datasets, possibly with a large number of event types.

## 2 Notation and Model Formulation

### 2.1 Preliminaries

An event dataset is denoted $D = \{(l_i, t_i)\}_{i=1}^{N}$, where $t_i$ is the occurrence time of the $i^{th}$ event, $t_i \in \mathbb{R}^+$, and $l_i$ is an event label/type belonging to a finite alphabet $\mathcal{L}$ with cardinality $|\mathcal{L}| = M$. We assume a temporally ordered dataset, $t_i \leq t_j$ for $i < j$, with initial time $t_0 = 0 \leq t_1$ and end time $t_{N+1} = T \geq t_N$, where $T$ is the total time period. Figure 1(a) shows an example event dataset with $N = 7$ events from the event label set $\mathcal{L} = \{A, B, C\}$ over $T = 20$ days.

In this paper, we propose learning algorithms that are data-driven; specifically, we will rely on inter-event times between event labels in the dataset. We denote the set of times from the most recent occurrence of $Z$, if $Z$ has occurred, to every occurrence of $X$ ($Z \neq X$) as $\{\hat{t}_{zx}\}$. We use $\{\hat{t}_{zz}\}$ to denote inter-event times between $Z$ occurrences, including the time from the last occurrence of $Z$ to the final time $T$. In the Figure 1(a) example, $\{\hat{t}_{ac}\} = \{2, 8\}$, $\{\hat{t}_{bc}\} = \{1, 7\}$ and $\{\hat{t}_{bb}\} = \{3, 7, 7\}$.

## 2.2 PGEM Formulation

An event dataset can be modeled using marked point processes, whose parameters are conditional intensity functions; in the most general case, the conditional intensity for event label $X$ is a function of the entire history, $\lambda_x(t|h_t)$, where $h_t$ includes all events up to time $t$, $h_t = \{(l_i, t_i) : t_i \leq t\}$. We use lower case $x$ wherever we refer to label $X$ in subscripts or parentheses. A graphical representation of a marked point process can help specify the historical dependence. For graph $\mathcal{G} = (\mathcal{L}, \mathcal{E})$ where nodes correspond to event labels, the conditional intensity for label $X$ depends only on historical occurrences of its parent event labels, therefore $\lambda_x(t|h_t) = \lambda_x(t|[h(\mathbf{U})]_{\mathbf{t}})$, where $\mathbf{U}$ are parents of node $X$ in $\mathcal{G}$ and $[h(\mathbf{U})]_{\mathbf{t}}$ is the history restricted to event labels in set $\mathbf{U}$ [Gunawardana and Meek, 2016]. We refer to nodes and event labels interchangeably.

A proximal graphical model $\mathcal{M}$ consists of a graph along with a set of (time) windows and conditional intensity parameters, $\mathcal{M} = \{\mathcal{G}, \mathcal{W}, \Lambda\}$. There is a window for every edge in the graph, $\mathcal{W} = \{w_x : \forall X \in \mathcal{L}\}$, where $w_x = \{w_{zx} : \forall Z \in \mathbf{U}\}$ denotes the set of all windows corresponding to incoming edges from $X$'s parents $\mathbf{U}$. $\Lambda = \{\lambda_{x|\mathbf{u}}^{w_x} : \forall X \in \mathcal{L}\}$ is the set of all conditional intensity parameters. For node $X$, there is a parameter for every instantiation $\mathbf{u}$ of its parent occurrences, depending on whether a parent event label has occurred in its window; thus there are $2^{|\mathbf{U}|}$ parameters for $X$, making the PGEM parametrization analogous to a Bayesian network with binary variables. To avoid notational clutter, we will hide the window superscript for conditional intensities. Figure 1(b) provides an illustrative PGEM graph along with the windows. In this example, parameter $\lambda_{c|a\bar{b}}$ signifies the rate at which $C$ occurs at any time $t$ given that $A$ has occurred at least once in the interval $[t - w_{ac}, t)$ and that $B$ has not occurred in $[t - w_{bc}, t)$.

# 3 Learning PGEMs

The learning problem is as follows: given an event dataset $D$, estimate a PGEM $\mathcal{M} = \{\mathcal{G}, \mathcal{W}, \Lambda\}$, i.e. parents, windows, and conditional intensity parameters for each event label. In this section, we first discuss learning windows and parameters for a node given its parents, and then present some theoretical results to help with parent search. We end the section with practical heuristic algorithms.

## 3.1 Learning Windows

When the parents $\mathbf{U}$ of all nodes $X$ are known, the log likelihood of an event dataset given a PGEM can be written in terms of the summary statistics of counts and durations in the data and the conditional intensity rates of the PGEM:

$$\mathrm{logL}(D) = \sum_X \sum_{\mathbf{u}} \left(-\lambda_{x|\mathbf{u}} D(\mathbf{u}) + N(x; \mathbf{u}) \ln(\lambda_{x|\mathbf{u}})\right), \tag{1}$$

where $N(x; \mathbf{u})$ is the number of times that $X$ is observed in the dataset and that the condition $\mathbf{u}$ (from $2^{|\mathbf{U}|}$ possible parental combinations) is true in the relevant preceding windows, and $D(\mathbf{u})$ is the duration over the entire time period where the condition $\mathbf{u}$ is true. Formally, $N(x; \mathbf{u}) = \sum_{i=1}^{N} I(l_i = X) I_{\mathbf{u}}^{w_x}(t_i)$ and $D(\mathbf{u}) = \sum_{i=1}^{N+1} \int_{t_{i-1}}^{t_i} I_{\mathbf{u}}^{w_x}(t) dt$, where $I_{\mathbf{u}}^{w_x}(t)$ is an indicator for whether $\mathbf{u}$ is true at time $t$ as a function of the relevant windows $w_x$. Note that we have hidden the dependence of the summary statistics on windows $w_x$ for notational simplicity.

From Equation 1, it is easy to see that the maximum likelihood estimates (MLEs) of the conditional intensity rates are $\hat{\lambda}_{x|\mathbf{u}} = \frac{N(x;\mathbf{u})}{D(\mathbf{u})}$. The following theorem uses this to provide a high-level recipe for finding optimal windows for a node given its parents. $N(x)$ denotes counts of event label $X$ in the data.

**Theorem 1.** *The log likelihood maximizing windows for a node $X$ with parents $\mathbf{U}$ are those that maximize the KL divergence between a count-based distribution with probabilities $\frac{N(x;\mathbf{u})}{N(x)}$ and a duration-based distribution with probabilities $\frac{D(\mathbf{u})}{T}$.*

Note that for each time $t \in [0, T]$, there is some one parental state $\mathbf{u}(h_t, w_x)$ that is active. Since the number of such parental states over $[0, T]$ is finite (upper bounded by $\min(2^{|\mathbf{U}|}, 2N)$ and further limited by what the data $D$ and windows $w_x$ allow), this leads to a finite partition of $[0, T]$. Each

member in this partition corresponds to some parental state $\mathbf{u}$, and in general, it is a union of a collection of non-intersecting half-open or closed time intervals that are subsets of $[0, T]$. Each member thus has a net total duration, which sums to $T$ across the above partition, and similarly a net total count of the number of arrivals of type $X$. As such, $w_x$ taken with $D$ is equivalent to two finite distributions (histograms) whose support is over the above set of partition members, one each for counts and the durations. The above theorem observes that the optimal $w_x$ is one where the count histogram across the partition members maximally differs from the corresponding duration histogram, as per KL divergence. (All proofs are in the supplementary section.) In informal terms, the windows $w_x$ that lead to MLE estimates for conditional intensities are the ones where the summary statistics of empirical arrival rates differ maximally across the above parental state partition.

The challenge with applying Theorem 1 to the practical issue of finding the optimal windows is that this is in general a difficult combinatorial optimization problem with a non-linear objective function. Figure 1(c) displays the shape of the log likelihood function for node $C$ as a function of windows from its parents $A$ and $B$ in the PGEM from Figure 1(b). Note that the maximization over regionally convex areas results in several local maxima.

Next we provide an upper bound on the optimal window from parent $Z$ to node $X$ regardless of other considerations.

**Theorem 2.** *The log likelihood maximizing window $w_{zx}$ from parent $Z$ to a node $X$ is upper bounded by $\max\{\hat{t}_{zz}\}$, where $\{\hat{t}\}$ denotes inter-event times, which is also taken to include the inter-event time between the last arrival of $Z$ and $T$ (end of the horizon).*

The following theorem shows that when a node has a single parent, one can discover a small number of local maxima from the inter-event times in the data, thereby easily computing the global maximum by exhaustively comparing all local maxima.

**Theorem 3.** *For a node $X$ with a single parent $Z$, the log likelihood maximizing window $w_{zx}$ either belongs to or is a left limit of a window in the candidate set $W^* = \{\hat{t}_{zx}\} \cup \max\{\hat{t}_{zz}\}$, where $\{\hat{t}\}$ denotes inter-event times.*

In the proof of Theorem 3, we show that the candidate window set arises primarily because the counts $N(x; z)$ only change at the inter-event times $\{\hat{t}_{zx}\}$. Note that the counts are step functions and therefore discontinuous at the jump points; this is the reason why the optimal window can be a left limit of an element in $W^*$. We deal with this practically by searching over both $W^*$ as well as $W^* - \epsilon$ ($\epsilon = 0.001$ is chosen for all experiments). Next, we show that the optimal window from parent $Z$ to node $X$ given other parents and their windows also belongs to a grid, albeit a finer one than in Theorem 3.

**Theorem 4.** *For a node $X$ and parent(s) $\mathbf{Y}$, the log likelihood maximizing window for a new parent $Z$, $w_{zx}$, given the windows corresponding to nodes from $\mathbf{Y}$ to $X$, either belongs to or is a left limit of a window in the candidate set $W^* = \{\hat{t}_{zx}\} \cup \hat{C}_{\mathbf{y},z}$, where $\{\hat{t}\}$ denotes inter-event times and $\hat{C}_{\mathbf{y},z}$ are change points across the set of piecewise linear functions $D(\mathbf{y}, z)$ (multiple functions, due to multiple parental state combinations) that are obtainable from Algorithm 1.*

$\hat{C}_{\mathbf{y},z}$ captures all the change points that are pertinent to any of the functions $D(\mathbf{y}, z)$ when a window $w$ is varied over $[0, \bar{W}]$, where $\bar{W} = \max\{\hat{t}_{zz}\}$ is an upper bound on the optimal $w$ (Theorem 2).

We will use the above two theorems in our heuristics for finding the optimal windows and parameters given a parent set.

### 3.2 Optimal Parent Set Search

In the literature on Bayesian networks, various scores such as Akaike information criterion (AIC), Bayesian information criterion (BIC), and Bayesian Dirichlet equivalent uniform (BDeu) are used to learn the graphical structure from data. These scores can be viewed as combining the log likelihood of the data with a term that penalizes the complexity of the model. Prior work has shown that these criteria enjoy some properties that can help parent search by eliminating non-optimal parent sets quickly, reducing the search space size and speeding up learning [Teyssier and Koller, 2005, Campos and Ji, 2011]. Here we provide similar theoretical results on parent set search in PGEMs. We use the

---

**Algorithm 1** Change points in $w$ across all of piece-wise linear functions $D(\mathbf{y}, z)$

---

**Inputs:** Dataset $D$, Labels $X, Z$, given parent set $\mathbf{Y}$ with windows $w_{yx}, y \in \mathbf{Y}$
**Outputs:** $\hat{C}_{\mathbf{y},z}$, initialized to $\emptyset$
**Initialize:** $S = \{[t_{z,i}, t_{z,i+1}]\}_{i=1}^{N(z)}$, ordered inter-arrival intervals of $Z$, with $t_{z,N(z)+1} = T$.
**for** all $y$ in $\mathbf{Y}$ **do**
    **for** all $s_i = [t_{z,i}, t_{z,i+1}]$ in $S$ **do**
        Let: $\mathtt{cl}(s_i) = \max_k \{t_{y,k} | t_{y,k} < t_{z,i}\}$
        Let: $\mathtt{in}(s_i) = \{t_{y,k} | t_{y,k} \in (t_{z,i}, t_{z,i+1})\}$
        $\hat{C}_{\mathbf{y},z} = \hat{C}_{\mathbf{y},z} \cup$ **changepoints**$(s_i, \mathtt{cl}(s_i), \mathtt{in}(s_i), y)$
$\hat{C}_{\mathbf{y},z} = \hat{C}_{\mathbf{y},z} \cup \{\hat{t}_{zz}^{(k)}\}$, i.e. add the set of the $N(z)$ ascending order statistics of label $Z$ inter-event times $\{\hat{t}_{zz}\}$, including the inter-event time between the last arrival of $Z$ and $T$

---

**changepoints**$(s_i, \mathtt{cl}(s_i), \mathtt{in}(s_i), y)$:
**Initialize:** Stack $\sigma = \emptyset, \mathcal{C} = \emptyset$
**if** $(\mathtt{cl}(s_i) + w_{yz}) < t_{z,i+1}$ **then**
    **if** $(\mathtt{cl}(s_i) + w_{yz}) \in (t_{z,i}, t_{z,i+1})$ **then**
        $\sigma$.push$(\mathtt{cl}(s_i) + w_{yz} - t_{z,i})$
    **for** all $t$ in $\mathtt{in}(s_i)$ **do**
        **if** $\sigma$ not empty **then**
            tail = $\sigma$.top
        **else**
            tail = -1
        **if** $t \leq$ tail **then**
            $\sigma$.pop
        **else**
            $\sigma$.push$(t - t_{z,i})$
        **if** $(t + w_{yx} > t_{z,i+1})$ **then**
            break
        **else**
            $\sigma$.push$(t + w_{yz} - t_{z,i})$
    $\mathcal{C} = \mathtt{set}(\sigma)$
return $\mathcal{C}$

---

BIC score in our experiments, defined for a PGEM as:

$$\text{BIC}(D) = \text{logL}(D) - \ln(T) \sum_X 2^{|\mathbf{U}|}. \tag{2}$$

First, we state a simple way to discard parent sets for a node in a PGEM, as used in Teyssier and Koller [2005], Campos and Ji [2011].

**Lemma 5.** *Let $X$ be an arbitrary node of $\mathcal{G}$, a candidate PGEM graph where the parent set of $X$ is $\mathbf{U}'$. If $\mathbf{U} \subset \mathbf{U}'$ such that $s_X(\mathbf{U}) > s_X(\mathbf{U}')$, where $s$ is BIC, AIC, BDeu or a derived scoring criteria, then $\mathbf{U}'$ is not the parent set of $X$ in the optimal PGEM graph $\mathcal{G}^*$.*

While Lemma 5 provides a way to eliminate low scoring structures locally, one still needs to compute the scores of all possible parent sets and then remove the redundant ones. The search still requires $M \times 2^M$ asymptotic score computation and the same complexity for parent score storage, although the space can be reduced after applying Lemma 5. We present the following two lemmas to reduce some of these computations. We focus on the BIC score but similar results should hold for other scores. Since BIC is decomposable, the local BIC score for node $X$ can be expressed as $s_X(\mathbf{U}) = L_X(\mathbf{U}) - t_X(\mathbf{U})$, where $L_X(\mathbf{U})$ is the log likelihood term and $t_X(\mathbf{U})$ is the structure penalty term, $t_X(\mathbf{U}) = \ln T \cdot 2^{\mathbf{U}}$.

**Theorem 6.** *Using BIC as the score function, suppose that $X$ and $\mathbf{U}$ are such that $2^{|\mathbf{U}|} > \frac{N(x)(1 - \ln N(x))}{\ln T} + N(x)$, where $2^{|\mathbf{U}|}$ is the total size of all possible parent combinations, $N(x)$ is the total count of $X$ in the data and $T$ is the maximal time horizon. If $\mathbf{U}'$ is a proper superset of $\mathbf{U}$, then $\mathbf{U}'$ is not the parent set of $X$ in the optimal PGEM graph.*

---

**Algorithm 2** Forward Backward Search

---

**Inputs:** Event label $X$, event dataset $D$
**Outputs:** Parents $\mathbf{U}$, windows $w_x$, lambdas $\lambda_{x|\mathbf{u}}$, score such as BIC

---

**Forward Search:** Initialize $\mathbf{U} = \emptyset$; $S = -\infty$
**while** score cannot be improved or no more parents can be added **do**
    **for** all $Z$ not in $\mathbf{U}$ **do**
        Find all optimal windows and $\lambda$s with $Z$ added to $\mathbf{U}$ and corresponding score $S(\mathbf{U} \cup Z)$
    **if** $\max_Z \{S(\mathbf{U} \cup Z)\} > S$ **then**
        Add $Z$ to $\mathbf{U}$, $S = \max_Z \{S(\mathbf{U} \cup Z)\}$

---

**Backward Search:** Start with parent set $\mathbf{U}$ and $S$ from forward search
**while** score cannot be improved or $\mathbf{U} = \emptyset$ **do**
    **for** all $Z$ in $\mathbf{U}$ **do**
        Find all optimal windows and $\lambda$s with $Z$ removed from $\mathbf{U}$ and corresponding score $S(\mathbf{U} \backslash Z)$
    **if** $\max_Z \{S(\mathbf{U} \backslash Z)\} > S$ **then**
        Remove $Z$ from $\mathbf{U}$, $S = \max_Z \{S(\mathbf{U} \backslash Z)\}$

---

**Corollary 7.** *Using BIC as the score function, the optimal graph $\mathcal{G}^*$ has at most $O(\log_2 N(x))$ parents for each node $X$.*

Theorem 6 and Corollary 7 ensure that we only need to compute $O(\sum_{k=0}^{\lceil \log_2 N(x) \rceil} \binom{M-1}{k})$ elements for each variable $X$.

The next theorem does not directly improve the theoretical size bound of the parent set size that is achieved by Corollary 7, but it helps in practice as it is applicable to cases where Theorem 6 is not, implying even fewer parent sets need to be tested.

**Theorem 8.** *Using BIC as score function $s$, let $X$ be a node with two possible parent sets $\mathbf{U} \subset \mathbf{U}'$ such that $t_X(\mathbf{U}') + s_X(\mathbf{U}) > 0$. Then $\mathbf{U}'$ and all its supersets $\mathbf{U}'' \supset \mathbf{U}'$ are not optimal parent sets for $X$ in the optimal PGEM graph.*

Hence, Theorem 8 can be used to discard additional parents sets without computing its local scores. Every time the score of a parent set $\mathbf{U}$ of $X$ is about to be computed, we can take the best score of any its subsets and test it against the theorem. If the condition applies, we can safely discard $\mathbf{U}$ and all its supersets. To summarize, we would need to build all possible parent sets up to $O(\log_2 N(x))$ for each $X$ and then use Theorem 8 and then Lemma 5 to test the optimal parent set.

### 3.3 A Forward-Backward Search Algorithm

We propose a forward-backward search (FBS) algorithm to learn the structure of a PGEM as shown in Algorithm 2. Since a PGEM can include cycles, there are no acyclicity constraints like in Bayesian networks, therefore we can run Algorithm 2 on each node/label $X$ separately. This local learning approach is similar to local learning in Bayesian networks [Gao and Wei, 2018] but can contain cycles.

Given an event data set $D$ and a target label $X$, FBS first initializes the parent set $\mathbf{U}$ to be empty. At each step of a forward search, FBS iteratively chooses a parent candidate $Z$ that is not in $\mathbf{U}$, and find the best window and rates $\lambda$ that maximize the score $S(\mathbf{U} \cup Z)$ with parent set $\mathbf{U} \cup Z$ (as discussed in Section 3.1). If the maximized $S(\mathbf{U} \cup Z)$ is better than the current best score $S$, then FBS chooses to add $Z$ to $\mathbf{U}$ and update $S$. It runs until all variables have been tested or no parent set would improve the score (as discussed in Section 3.2). Then during the backward search step, FBS iteratively tests if each variable $Z$ in $\mathbf{U}$ can be removed, i.e. if the removed set $\mathbf{U} \setminus Z$ would give a better score. If so, $Z$ would be removed from $\mathbf{U}$. Backward search runs until score $S$ cannot be improved or $\mathbf{U}$ becomes empty.

With the optimal parent set search with bounded sizes and determination of optimal windows and conditional intensity rates given a graph, one can show the soundness and completeness of Algorithm 2 under mild assumptions. Gunawardana and Meek [2016] show that backward and forward search with

BIC scores is sound and complete for a family of GEMs. Assuming that the underlying distribution can be captured uniquely by a PGEM model, then since PGEMs can be considered a sub-class of this family and Algorithm 2 is a similar forward-and-backward search, soundness and completeness applies in this instance as well.

**Theorem 9.** *Under the large sample limit and no detailed balance assumptions [Gunawardana and Meek, 2016], Algorithm 2 is sound and complete.*

Jointly optimizing the windows for multiple parents simultaneously is a hard problem in general. We instead realize two efficient heuristics based on the above FBS procedure, namely FBS-IW and FBS-CW. In FBS-IW, we independently optimize the window for each parent relative to label $X$, using the finite characterization of single-parent optimal windows presented in Theorem 3. After each individual parent's window has been independently optimized, we compute the corresponding finite partition of $[0, T]$ in terms of parental states, and use the sufficient statistics in each partition member to estimate the corresponding conditional intensity rates. In FBS-CW, we appeal to Theorem 4 and realize a block coordinate ascent strategy (over parent labels) for optimizing the windows. For each parent that is added in the forward search, we optimize its window while keeping all the other existing parents fixed at their current windows. The rate estimation is then as described above for FBS-IW. We add parents in the forward search if there is a score improvement based on the new windows and rates. For the backward search, we delete a parent, retain existing windows for remaining parents and only recompute the intensity rates in both FBS-IW and FBS-CW.

**Theorem 10.** *If all event labels occur in the dataset in similar proportions, the worst case complexity of the FBS-IW and FBS-CW algorithms are $O(N^2 + M^3 N)$ and $O(M^3 N^2)$ respectively.*

## 4 Experiments

We consider 2 baselines for our experiments. A superposition of independent Poisson (SIP) arrivals is a weak baseline that treats every event label as an independent Poisson process and is equivalent to a PGEM without edges. We also test the CPCIM algorithm [Parikh et al., 2012], shown to be an improved version over piecewise constant intensity model (PCIM) [Gunawardana et al., 2011] and other variants, to compare the performance of the proposed algorithm. For CPCIM, we used the following hyper-parameters. The conjugate prior for conditional intensity has two parameters, the pseudo-count $\alpha$ and pseudo-duration $\beta$ for each label. We used the same values for all labels, by computing a ratio $\rho$ of the total number of all arrivals over all labels to the total duration for all labels (the product of the number of labels and the horizon $T$ under consideration). This ratio provides an empirically based estimate of the arrival rate. We ran experiments using $\alpha = K\rho$, $\beta = K$, for various values of $K = 10, 20, \ldots$, where higher values of $K$ correspondingly increase the influence of the prior on the results. Experimental results presented in this section are for $K = 20$. The structural prior $\kappa$ was fixed at 0.1 [Gunawardana et al., 2011]. We also experimented with MFPP [Weiss and Page, 2013] which is based on random forests, but we observed high sensitivity to forest parameters as well as randomness in the optimized log likelihood values which went to negative infinity in many runs. We therefore present comparisons with only SIP and CPCIM in the experi ments. Both PGEM learning algorithms use $\epsilon = 0.001$ to search for left limiting points.

### 4.1 Synthetic Datasets

We generate PGEMs for a label set $\mathcal{L}$ of size $M$ through the following process. For each node, we select the number of its parents $K$ uniformly from the parameters $K_{min} \geq 0, \cdots, K_{max} \leq M$ in integer increments; a random subset of size $K$ from $\mathcal{L}$ is then chosen as its parent set. We generate windows for each edge uniformly from $w_{min}$ to $w_{max}$ in increments of $\Delta w$. For the conditional intensity rates, we assume that each node's parent either has a multiplicative amplification or damping rate beyond a baseline rate of $r/M$ ($r = 1$ implies an overall rate of one label per time period in the dataset). Nodes that always increase occurrence rate for their children are obtained by randomly choosing a subset $\mathcal{L}_A$ of size $K_A$ from $\mathcal{L}$. Nodes in the sets $\mathcal{L}_A$ and $\mathcal{L} \backslash \mathcal{L}_A$ have an amplification and damping rate of $\gamma_A$ and $\gamma_D$ respectively.

Figure 2 compares models using 6 PGEMs generated from the afore-mentioned process; the top and bottom rows have PGEMs with $M = 5$ and $M = 10$ labels respectively. Other details about the model generating parameters are described in the supplementary material. For each model, we generated 10 event datasets over $T = 1000$ days (around 3 years) from a synthetic PGEM generator. Windows

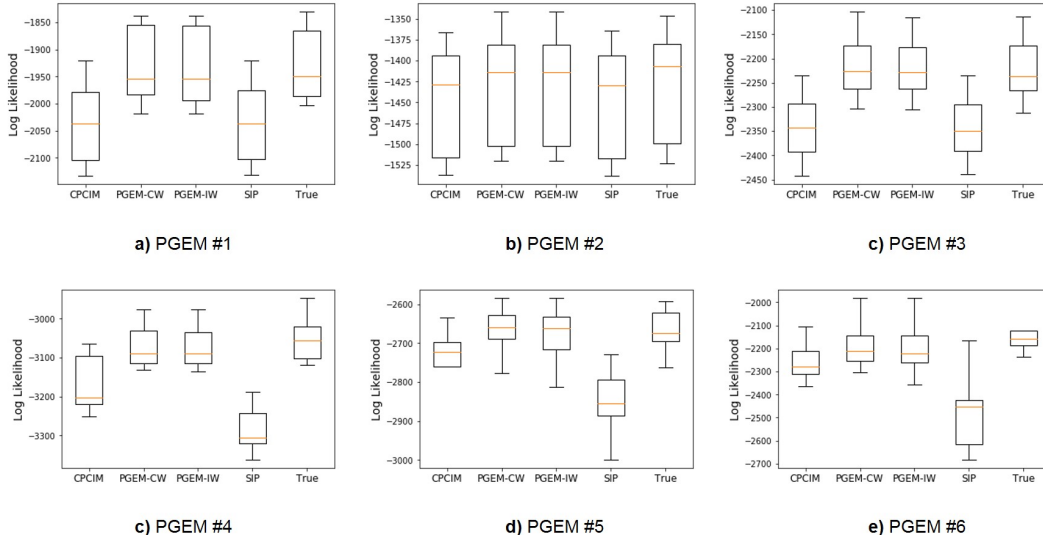

Figure 2: Model comparisons with 10 synthetic event datasets generated from 6 PGEMs. The top and bottom rows have PGEMs with $M = 5$ and $M = 10$ labels respectively. Both PGEM learning algorithms (FBS-IW and -CW) are compared with baseline models SIP and CPCIM as well as the true model.

Table 1: Log likelihood of models for experiments on the books dataset

| Dataset | SIP | PGEM | CPCIM |
|---|---|---|---|
| Leviathan ($M = 10$) | -19432 | **-18870** | -19237 |
| Leviathan ($M = 20$) | -36398 | **-35179** | -36055 |
| Bible ($M = 10$) | -76097 | **-72013** | -72801 |
| Bible ($M = 20$) | -147706 | **-138190** | -140327 |

were chosen to range from between a fortnight to 2 months. For CPCIM, we used intervals of the form $[t - t^*, t)$ as basis functions, where $t^* \in \{1, 2, 3, 4, 5, 6, 7, 15, 30, 45, 60, 75, 90, 180\}$. The boxplots indicate that the PGEM learning algorithms beat the baselines and come close to matching the log likelihood of the true model on the datasets. We observed in these and other experiments that the PGEM learning algorithms perform comparably; we therefore restrict our attention to the more efficient FBS-IW algorithm in subsequent experiments.

## 4.2 Real Datasets

**Books.** We consider two books from the SPMF data mining library [Fournier-Viger et al., 2014]: Leviathan, a book by Thomas Hobbes from the 1600s, and the Bible. We ignore the 100 most frequent words to remove stop-words and only retain the next most frequent $M$ words; this provides us with large event datasets where every word in scope is an event label and its index in the book is the occurrence time. For the Bible with $M = 20$, there are $N = 19009$ words. Table 1 shows that PGEM has greater log likelihood than the baselines on the four datasets considered. For CPCIM, we used intervals of the form $[t - t^*, t)$ as basis functions, where $t^* \in \{25, 50, 100, 200, 300, 400, 500, 1000, 5000\}$. These datasets revealed to us how challenging it could be to identify basis functions, thereby reinforcing the benefits of PGEMs.

From Table 1, we see that PGEM outperforms both SIP and CPCIM consistently on the book datasets, while CPCIM is better than SIP. PGEM achieves the best result on all 4 datasets, with the smallest margin of $400$ in LL and up to 2000 over CPCIM.

**ICEWS.** We consider the Integrated Crisis Early Warning System (ICEWS) political relational event dataset [O'Brien, 2010], where events take the form '*who* does *what* to *whom*', i.e. an event $z$ involves a source actor $a_z$ performing an action/verb $v_z$ on a target actor $a'_z$, denoted $z = (a_z, v_z, a'_z)$. In ICEWS, actors and actions come from the Conflict and Mediation Event Observations (CAMEO)

Table 2: Log likelihood of models for experiments on the ICEWS dataset

| Dataset | SIP | PGEM | CPCIM |
|---|---|---|---|
| Argentina | -11915 | **-8412** | -10631 |
| Brazil | -14289 | **-8856** | -11706 |
| Colombia | -4621 | **-2965** | -3557 |
| Mexico | -7895 | -6011 | **-5676** |
| Venezuela | -8922 | **-5454** | -6757 |

ontology [Gerner et al., 2002]. Actors in this ontology could either be associated with generic actor roles and organizations (ex: *Police (Brazil)*) or they could be specific people (ex: *Hugo Chavez*). Actions in the CAMEO framework are hierarchically organized into 20 high-level base coded actions that range 1-20. For our experiment, we restricted attention to five countries, namely, Brazil, Argentina, Venezuela, Mexico and Colombia over a four year time period, Jan 1 2012 to Dec 31, 2015. We included only 5 types of actors, namely, Police, Citizen, Government, Head of Government and Protester, normalizing for actual heads of governments (i.e. mapping Hugo Chavez to Head of Government (Venezuela) for e.g.). We considered 5 types of actions, namely, Neutral [1-2], Verbal cooperation [3-5], Material cooperation [6-8], Verbal conflict [9-13] and Material conflict [14-20], where the numbers in the brackets show how the action categories map to the CAMEO codes. For CPCIM, we used intervals of the form $[t - t^*, t)$ as basis functions, where $t^* \in \{7, 15, 30, 45, 60, 75, 90, 180\}$. From Table 2, we see that PGEM outperforms both SIP and CPCIM on 4 out of 5 countries, while CPCIM is better than PGEM for Mexico.

## 5 Conclusions

In this paper, we introduce a novel model for event datasets – proximal graphical event models – with the following major contributions: 1) We study the optimal window size in PGEMs and conduct theoretical analysis; 2) We derive efficient parent set size bounds in PGEMs for usage in structure learning algorithms; 3) We propose a forward-backward search algorithm, with two efficient heuristics, to learn the structure and parameters of PGEMs; 4) We demonstrate PGEM's superior modeling power on multiple synthetic and real datasets. PGEMs do not require careful tuning of many hyper-parameters compared to existing methods, making them useful along with interpretable. In practice, given the underlying parametric assumptions of a PGEM and the proposed heuristic approach to obtaining windows, the learning approach could potentially mis-characterize causal/acausal relationships between event types in more complex underlying distributions. Nevertheless, we believe that PGEMs are readily suitable for many real world applications.

### Acknowledgments

We thank Nicholas Mattei and Karthikeyan Shanmugam for helpful discussions, Christian Shelton for help with the CPCIM code, and three anonymous reviewers for their valuable feedback.

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
