[Supplementary Material]

# 6 Supplementary Material for Proximal Graphical Event Models

## 6.1 Theorem Proofs

### 6.1.1 Proof for Theorem 1

The log likelihood of the data given a PGEM equals:

$$\sum_{\mathbf{u}} N(x; \mathbf{u}) \ln \frac{N(x; \mathbf{u})}{D(\mathbf{u})} - \sum_{\mathbf{u}} N(x; \mathbf{u}) = \quad (\Leftarrow \text{Replacing } \hat{\lambda}_{x|\mathbf{u}} = \frac{N(x; \mathbf{u})}{D(\mathbf{u})})$$

$$\sum_{\mathbf{u}} N(x; \mathbf{u}) \ln \frac{N(x; \mathbf{u})}{D(\mathbf{u})} - N(x) = \quad (\Leftarrow \text{Since } \sum_{\mathbf{u}} N(x; \mathbf{u}) = N(x))$$

$$-N(x) + N(x) D_{KL}((N(x; \mathbf{u})/N(x)) || (D(\mathbf{u})/T))$$
$$-N(x) \ln N(x) + N(x) \ln T$$

The last step is from the definition of KL divergence and after re-arranging constants. Maximizing the log likelihood is therefore equivalent to maximizing the KL divergence in the final term.

### 6.1.2 Proof for Theorem 2

As presented in the discussion that immediately follows the presentation of Theorem 1 in the main paper, any choice of windows $w_x$ for node $X$ leads to a finite partition of $[0, T]$. Each member in this partition corresponds to some parental state $\mathbf{u}$, and in general, it is a union of a collection of non-intersecting half-open or closed time intervals that are subsets of $[0, T]$. Each member thus has a net total duration, which sums to $T$ across the above partition, and similarly a net total count of the number of arrivals of type $X$. (Note that each parental state $\mathbf{u}$ has a binary bit (active/absent) for each parent under consideration. The marginal component of this parental state corresponding to any specific parent simply refers to whether that specific parent is active or absent).

Consider any pair-wise window, say $w_{zx}$ for any parent $Z$ and child $X$. As this window increases, it changes the above partition in terms of which subsets of $[0, T]$ correspond to which parental state. Recall that $\{\hat{t}_{zz}\}$ and $\{\hat{t}_{zx}\}$ denote inter-event times, and that the former includes time from the last occurrence of $Z$ to the final time $T$. Let $\hat{t}_{zz}^{(k)}$ denote the $N(z)$ ascending order statistics of the inter-event times $\{\hat{t}_{zz}\}$, including the inter-event time between the last arrival of $Z$ and $T$. We argue that when $w_{zx}$ gets bigger than $\hat{t}_{zz}^{(N(z))}$, i.e. bigger than the biggest such inter-event time, it means all times in $[t_{z,1}, T]$ correspond to parent states whose marginal component for parent $Z$ will be active. Here $t_{z,1}$ denotes the time of first occurrence of $Z$. (Note that all times in the complement, namely all times in $[0, t_{z,1})$, correspond to parent states whose marginal component for parent $Z$ is inactive by definition in PGEM, regardless of the value of $w_{zx}$). Increasing $w_{zx}$ beyond $\hat{t}_{zz}^{(N(z))}$ cannot possibly change this marginal component in the resulting partition, since the entire relevant subset of the horizon is already marked active from the perspective of component $Z$. This establishes an upper bound for any $w_{zx}$. We also note that $T$ is a trivial upper bound regardless of any further argument since the horizon under consideration, and hence all arrival epochs under consideration as well as any inter-event windows of influence, are bounded by $T$. Theorem 2 provides a tighter upper bound that we exploit in our algorithms.

### 6.1.3 Proof for Theorems 3 and 4

Both Theorem 3 and Theorem 4 pertain to univariate optimization of the window between parent $Z$ and node $X$. The optimal window lies in an interval $[0, \bar{W}]$ where $\bar{W} = \max\{\hat{t}_{zz}\} < T$ (see Theorem 2).

For node $X$ with parent $Z$, let $w = w_{zx}$ denote the univariate window that needs to be optimized. Let $\hat{t}_{zz}^{(k)}$ and $\hat{t}_{zx}^{(k)}$ denote the ascending order statistics of these inter-event times respectively.

From their respective definitions, note that the count $N(x; z)$ is a step function of the inter-event times $\{t_{zx}\}$, whereas the duration $D(z)$ is piecewise linear in the inter-event times $\{t_{zz}\}$. Formally, $N(x; z) = \sum_{k=1}^{N(x)} I(w \geq \hat{t}_{zx}^{(k)})$ and $D(z) = \sum_{k=1}^{N(z)} \min(w, \hat{t}_{zz}^{(k)})$.

The log likelihood for node $X$ with parent $Z$ can be written as:

$$logL_X = N(x;z)\ln\frac{N(x;z)}{D(z)} + N(x;\bar{z})\ln\frac{N(x;\bar{z})}{D(\bar{z})}$$

Consider candidate window $w$ between some $w_0 = \hat{t}_{zx}^{(k)}$ and $w' = \hat{t}_{zx}^{(k+1)}$. Note that the counts remain constant from $w_0$ to the left limit of $w'$. Therefore, for $w \in [w_0, w')$, maximizing the log likelihood is equivalent to minimizing:

$$g(w) = N(x;z)\ln f(w) + (N(x) - N(x;z))\ln(T - f(w)),$$

where $f(w) = D(z)$ is an increasing piecewise linear function and the count $N(x;z)$ is constant in the interval $[w_0, w')$. The derivative is:

$$g'(w) = f'(w)\left[\frac{N(x;z)}{f(w)} - \frac{(N(x) - N(x;z))}{(T - f(w))}\right]$$

This derivative is negative at $w = w_0$ if $f(w_0) > \frac{N(x;z)}{N(x)} \times T$. Since $f(w)$ is an increasing function, increasing the window will continue to keep the derivative negative, in which case the left limit of $w'$ is the window that minimizes $g(w)$. Alternatively, the gradient is positive at $w = w_0$ and either stays positive as $w$ is increased to $w'$ or switches sign when $f(w) = \frac{N(x;z)}{N(x)} \times T$. The optimal window in this interval must therefore either be $w_0$ or the left limit of $w'$.

The resulting candidate set $W^*$ follows from recognizing that $w = 0$ cannot be the optimal window and that the optimal window is either at a point or a left limit of where the counts change, as shown by the analysis above, or at the maximum window $\bar{W} = \max\{\hat{t}_{zz}\}$.

Theorem 4 involves the situation where node $X$ has parents $\mathbf{Y}$ and we are considering adding a new parent $Z$ in addition. The proof is similar, except that along with the candidate windows where the counts change, one also needs to consider the additional changepoints in the piecewise linear description in any of the various duration functions, each of which is perturbed by changing the univariate $w = w_{zx}$ under question. In this situation, the log likelihood for node $X$ is:

$$logL_X = \sum_{\mathbf{y}} N(x;\mathbf{y},z)\ln\frac{N(x;\mathbf{y},z)}{D(\mathbf{y},z)} + \sum_{\mathbf{y}} N(x;\mathbf{y},\bar{z})\ln\frac{N(x;\mathbf{y},\bar{z})}{D(\mathbf{y},\bar{z})}$$

As before, if we consider $w$ between some $w_0 = \hat{t}_{zx}^{(k)}$ and $w' = \hat{t}_{zx}^{(k+1)}$, then since the counts remain constant for $w \in [w_0, w')$, maximizing the log likelihood is equivalent to minimizing:

$$g(w) = \sum_{\mathbf{y}} N(x;\mathbf{y},z)\ln f_{\mathbf{y}}(w) + \sum_{\mathbf{y}} (N(x;\mathbf{y}) - N(x;\mathbf{y},z))\ln(D(\mathbf{y}) - f_{\mathbf{y}}(w)),$$

where $f_{\mathbf{y}}(w) = D(\mathbf{y},z)$ is a piecewise linear function in $w$. Performing similar analysis as conducted earlier, if we restrict attention to intervals between points where either the counts change, such as $w_0$ and $w'$, but also the points where the durations $D(\mathbf{y},z)$ change for all $\mathbf{y}$, then the optimal window must either lie at the beginning of the interval or is a left limit of the end of the interval. Algorithm 1 constructively identifies in linear time all such changepoints without trying to identify which of the several duration functions (upper bounded in count by $\min(2^{|\mathbf{U}|}, 2N)$) any specific changepoint corresponds to.

### 6.1.4 Proof for Lemma 5

*Proof.* The proofs are practically the same as in non-event DAG learning. Take a graph $\mathcal{G}$ that differs from $\mathcal{G}^*$ only on the parent set of $X$, where it has $\mathbf{U}$ instead of $\mathbf{U}'$. Then $s(\mathcal{G}) = \sum_{j\neq X} s_j(\mathbf{U}'_j) + s_X(\mathbf{U}) > \sum_{j\neq X} s_j(\mathbf{U}'_j) + s_X(\mathbf{U}') = s(\mathcal{G}')$. Since $\mathcal{G}'$ with $\mathbf{U}$ for $X$ has a subgraph $\mathcal{G}$ with a better score, $\mathbf{U}'$ is not the optimal parent set of $X$ in $\mathcal{G}^*$. $\qquad\square$

### 6.1.5 Proof for Theorem 6

*Proof.* Since $\mathbf{U}'$ is a proper superset, then $\mathbf{U}' \supseteq \mathbf{U} \cup X_e$ and $X_e \notin \mathbf{U}$. Because $\mathbf{U} \subset \mathbf{U}'$, $L_X(\mathbf{U}') \geq L_X(\mathbf{U})$ and $t_X(\mathbf{U}') > t_X(\mathbf{U})$. Then the difference in scores is $s_X(\mathbf{U}') - s_X(\mathbf{U})$, which equals:

$$\max_{\theta'} L_X(\mathbf{U}') - t_X(\mathbf{U}') - (\max_{\theta_X} L_X(\mathbf{U}) - t_X(\mathbf{U})) \leq \quad (\Leftarrow \text{from BIC definition})$$

$$-\max_{\theta_X} L_X(\mathbf{U}) - \mathbf{t_i}(\mathbf{U}') + \mathbf{t_X}(\mathbf{U}) = \quad (\Leftarrow \text{as } L_X(\mathbf{U}') < 0)$$

$$N(x) - \sum_{u=1}^{2^{|\mathbf{U}|}} N(x;\mathbf{u}) \ln \frac{N(x;\mathbf{u})}{D_{\mathbf{u}}} - t_X(\mathbf{U}') + t_X(\mathbf{U}) = \quad (\Leftarrow \text{by definition of} L_X(\mathbf{U}))$$

$$N(x) - N(x)\sum_{u=1}^{2^{|\mathbf{U}|}} \frac{N(x;\mathbf{u})}{N(x)} \ln \frac{N(x;\mathbf{u})/N(x)}{D_{\mathbf{u}}/T} - N(x)\ln N(x)+$$

$$N(x)\ln T - t_X(\mathbf{U}') + t_X(\mathbf{U}) = \quad (\Leftarrow \text{by adding zeros})$$

$$N(x) - N_X D_{kl}((N(x;\mathbf{u})/N(x))||(D_{\mathbf{u}}/T))$$

$$-N(x)\ln N(x) + N(x)\ln T - t_X(\mathbf{U}') + t_X(\mathbf{U}) \leq \quad (\Leftarrow \text{by KL-div definition})$$

$$N(x) - N(x)\ln N(x) + N(x)\ln T - (2^{|\mathbf{U}|}2^{|X_e|} - 2^{|\mathbf{U}|})\ln T = \quad (\Leftarrow \text{by KL-div property})$$

$$N(x) - N(x)\ln N(x) + N(x)\ln T - 2^{|\mathbf{U}|}(2^{|X_e|} - 1)\ln T \leq \quad (\Leftarrow \text{by factorization})$$

$$N(x) - N(x)\ln N(x) + N(x)\ln T - 2^{|\mathbf{U}|}\ln T$$

Finally, $N(x) - N(x)\ln N(x) + N(x)\ln T - 2^{|\mathbf{U}|}\ln T < 0$ if $N(x) - N(x)\ln N(x) + N(x)\ln T < 2^{|\mathbf{U}|}\ln T$, $\Rightarrow 2^{|\mathbf{U}|} > \frac{N(x)(1-\ln N(x))}{\ln T} + N(x)$. Hence, $s_X(\mathbf{U}') < s_X(\mathbf{U})$ if the condition holds, and Lemma 5 guarantees that $\mathbf{U}'$ cannot be the parent set of $X_i$ in the optimal structure. $\square$

### 6.1.6 Proof for Corollary 7

*Proof.* Assuming $T > 2$ and $N(x) > 2$, hence $\ln T > 1$ and $\ln N(x) > 1$, $\Rightarrow 1 - \ln N(x) < 0 \Rightarrow \frac{(1-\ln N(x))}{\ln T} + 1 < 1$, $\Rightarrow \frac{N(x)(1-\ln N(x))}{\ln T} + N(x) < N(x)$. Take a variable $X$ with a parent set $\mathbf{U}$ with exactly $\log_2 N(x)$ elements. Since each variable has two states, then $2^{|\mathbf{U}|} \geq N(x) > \frac{N(x)(1-\ln N(x))}{\ln T} + N(x)$. By Theorem 6, we know no proper superset of $\mathbf{U}$ can be optimal parent set, hence the maximal set size is bounded by $\log_2 N(x)$. $\square$

### 6.1.7 Proof for Theorem 8

*Proof.* The proof follows Theorem 4 of [Campos and Ji, 2011]. If $t_X(\mathbf{U}') + s_X(\mathbf{U}) > 0$, then $\Rightarrow -t_X(\mathbf{U}') - s_X(\mathbf{U}) < 0$. Therefore:

$$L_X(\mathbf{U}') - t_X(\mathbf{U}') - s_X(\mathbf{U}) < 0 \Rightarrow s_X(\mathbf{U}') - s_X(\mathbf{U}) < 0$$

.

Using Lemma 5, $\mathbf{U}'$ is not the optimal parent set for $X$. Similarly, the result follows for any $\mathbf{U}'' \supset \mathbf{U}'$ as $t_X(\mathbf{U}'') > t_X(\mathbf{U}')$. $\square$

### 6.1.8 Proof for Theorem 9

Theorem 7 and 9 in Gunawardana and Meek [2016] show consistency of the BIC score and forward-backward search in a more general GEM model, specifically the recursive timescale GEM (RTGEM). RTGEM relies on four operators on edges to guarantee the parameter and structure learning consistency: add edge, delete edge, split the time interval, or extend the time interval. If we assume the underlying true model is a PGEM, which only has one time interval (a recent/proximal time interval), then two of the four operators – the split and extend operators – are not needed. Hence, by using the add and delete edge operators, PGEM can learn the same models as RTGEM, assuming that PGEM learns the ground truth windows and conditional intensity functions. Therefore, the consistency of our algorithm directly follows Theorem 7 and 9 in Gunawardana and Meek [2016].

### 6.1.9 Proof for Theorem 10

Inter-event times can be obtained in time $O(MN)$. Also, one can divide the dataset into datasets with only pairs of event labels in $O(MN)$. For the FBS-IW algorithm, one can pre-compute optimal windows for all pairs $Z$ to $X$. If labels occur in around the same proportion, the sizes of the candidate window set as well as pairwise datasets are $O(N/M)$. Finding optimal windows for all pairs is therefore $M^2 * (N/M)^2 = O(N^2)$. In the forward search, when there are $K$ parents for node $X$, the dataset can be filtered (in $O(N)$) to a size of $O(KN/M)$. To compute the score for a node $X$ with a given graph and windows, one needs to compute the log likelihood from the summary statistics. This can be done in $O(K)$ time for each point in the dataset. In the worst case, all nodes are added, in time $O(\sum_{K=1}^{M} K^2 N/M) = O(M^2 N)$. Running this for all nodes is therefore $O(M^3 N)$. The backward search has similar time complexity.

For the FBS-CW algorithm, one cannot pre-compute windows; they are obtained conditionally, given existing parents and windows. Recall that for a node $X$ with $K$ parents, the dataset can be filtered (in $O(N)$) to a size of $O(KN/M)$; in this case, $O(KN/M)$ is also the size of the number of candidate windows. In the worst case, all nodes are added, and $O(K)$ work is done for all candidate windows and for all points in the dataset, in time $O(\sum_{K=1}^{M} K^3 N^2/M^2) = O(M^2 N^2)$. Running this for all nodes is therefore $O(M^3 N^2)$. The backward search is more efficient as windows are not re-computed.

## 6.2 PGEMs for Synthetic Datasets

The 6 PGEMs that generated the synthetic datasets were constructed using the following model building parameters:

**PGEM #1:**  Graph: $M = 5$, $K_{min} = 0$, $K_{max} = 5$; Windows: $w_{min} = 15$, $w_{max} = 30$, $\Delta w = 5$; Lambdas: $r = 1$, $K_A = 5$, $\mathcal{L}_A = \{2, 3, 4\}$, $\gamma_A = 2$, $\gamma_D = 0.25$.

**PGEM #2:**  Graph: $M = 5$, $K_{min} = 0$, $K_{max} = 5$; Windows: $w_{min} = 30$, $w_{max} = 60$, $\Delta w = 10$; lambdas: $r = 1$, $K_A = 5$, $\mathcal{L}_A = \{1, 3, 4\}$, $\gamma_A = 2$, $\gamma_D = 0.25$.

**PGEM #3:**  Graph: $M = 5$, $K_{min} = 0$, $K_{max} = 5$; Windows: $w_{min} = 15$, $w_{max} = 30$, $\Delta w = 5$; Lambdas: $r = 1$, $K_A = 5$, $\mathcal{L}_A = \{1, 2, 3\}$, $\gamma_A = 3$, $\gamma_D = 0.2$.

**PGEM #4:**  Graph: $M = 10$, $K_{min} = 0$, $K_{max} = 5$; Windows: $w_{min} = 15$, $w_{max} = 30$, $\Delta w = 5$; Lambdas: $r = 1$, $K_A = 5$, $\mathcal{L}_A = \{1, 2, 3, 5, 10\}$, $\gamma_A = 2$, $\gamma_D = 0.25$.

**PGEM #5:**  Graph: $M = 10$, $K_{min} = 0$, $K_{max} = 5$; Windows: $w_{min} = 30$, $w_{max} = 60$, $\Delta w = 10$; lambdas: $r = 1$, $K_A = 5$, $\mathcal{L}_A = \{2, 3, 5, 6, 9\}$, $\gamma_A = 2$, $\gamma_D = 0.25$.

**PGEM #6:**  Graph: $M = 10$, $K_{min} = 0$, $K_{max} = 5$; Windows: $w_{min} = 15$, $w_{max} = 30$, $\Delta w = 5$; Lambdas: $r = 1$, $K_A = 5$, $\mathcal{L}_A = \{2, 5, 6, 7, 9\}$, $\gamma_A = 3$, $\gamma_D = 0.2$.