[Reviews · NeurIPS 2018]

Reviewer 1



Proximal Graphical Event Models The paper describes proximal graphical event models (PGEM) to characterize relationships between events based on whether parent events of an event actually happen in a time window that preceded that event. The author suggest a framework for setting the optimal width of a window and a greedy structure-learning algorithm to learn a PGEM to maximize BIC. In addition, the authors evaluate their algorithm compared with two alternatives using several synthetic and real-world databases. Overall, the paper is well written, clear, original, convincing, and motivated. My main concern is related to the structure-learning approach that is incremental in a greedy fashion, where parents for each variable are learned conditioned on the variable own optimal window, and windows can have different sizes (durations) and overlap. Then there is a danger that the domain causal meaning will be missed out (say explaining a disease in the light of comorbidities and/or symptoms of comorbidities that have different causes). I would expect the authors to consider and discuss this issue. Major issues Although Lemmas 2-4 are intuitively proved, they invite formal proofs. I do not see major differences among the three graphs tested in Section 4.1, and thus I would expect the experimental evaluation to consider broader ranges of parameters for the synthesis of data. Minor issues References: Capitalize “Bayesian”, and keep a single format wrt to author names 125 – “ins” -> “is” 174 t_i(U) -> t_x(U) 196 “Since label X can be learned independently without concern of DAG violation” – needs clarification 197 “we can algorithm 2 on each X” – missing sentence that needs completion 233 – explain what is PCIM Supplementary material: 421 - Proof for Theorem 6 (not 8) 436 – missing ) Quality: good Clarity: very good Originality: very good Significance: high

Reviewer 2



This paper deals with event model learning from an event dataset, by considering that the probability of an event depends on the occurrence of some other event (the "parent" events) in the recent history. The Proximal graphical even model proposed in this paper is an extension of the Graphical event models of the literature, where the authors have added the explicit definition of the time window the parent events must occur to "generate" each target event. The authors proposes one learning algorithm decomposed in several steps : learning the windows for already known parents, then identifying the optimal parent set by a forward-backward algorithm. optimizing the windows for multiple parents simultaneously is one hard problem, so the authors proposed 2 heuristics, one simple one where each parent is treated independently, and another one where a block coordinate ascent strategy is used over the parent set. Experiments are performed with synthetic datasets where the more complex strategy doesn't seem to give better results than the first (independent) strategy. This last strategy is then used with real datasets and gives interesting results compared to competitive solutions. Some questions or comments: - why considering instantaneous events in your event datasets where process mining algorithms can deal with events defined in time windows. Is it possible to extend GEM and PGEM in this context ? - GEM deal with "transitions" between events, but how do you deal with the "initial" distribution of events ? - why considering the last time between an event and the end of the sequence in the set t_{xx} ??? in a "process" point of view, it should be the time between x and the "END" state ? - in your PGEM definition, you restrict yourself with the occurence of one parent event in only one time window, could you define the occurence of B with a presence of A in one time window and the absence of A in another one ??? (so considering that A could be present twice in the parent list of B, with 2 different time windows ?) - 3.1, equation 1. The log likelihood is defined for every value \lambda. \hat\lambda is the argmax of this log likehood rwrt \lambda. What your are defining in eqn 1 is the optimal value of the logL. - line 170 : typo : the computation still NEED? ...." - theorem 6 : is your assumption over 2^|U| realistic ? - line 197 : can RUN algo.2 ?

Reviewer 3



The main drawback of Graphical Event Models is that they require the user to specify a set of basis functions, which may requires hardworking in hyper-parameter tuning. The authors then propose a new proximal graphical event models (PGEM), which can learn such a set of basis functions. My main concern about the paper is the writing. (1) In section 2.2, the authors gives the definition of a proximal graphical model M. However, the authors did not give the associated distribution, and then directly goes to the log-likelihood in Section 3.1. For readers who are not familar with this topic it is hard to understand. (2) The authors uses abbreviations such as BIC, AIC and BD directly, for readers who does have background, it prevents readers to get the main idea in several sections. Maybe the authors should write a small section about these definitions. Minor problems: First page: (REF: Zhou et. al. 2013) should be fixed with correct citations.